# A co-anchoring strategy for the synthesis of polar bimodal polyethylene

Chen Zou[1,2], Quan Wang[1,2], Guifu Si [1] ✉ & Changle Chen [1] ✉

Since polar groups can poison the metal centers in catalysts, the incorporation of polar comonomers usually comes at the expense of catalytic activity and polymer molecular weight. In this contribution, we demonstrate polar bimodal polyethylene as a potential solution to this trade-off. The more-polar/more-branched low-molecular-weight fraction provides polarity and processability, while the less-polar/less-branched high-molecular-weight fraction provides mechanical and melt properties. To achieve high miscibility between these two fractions, three synthetic routes are investigated: mixtures of homogeneous catalysts, separately supported heterogeneous catalysts, and a co-anchoring strategy (CAS) to heterogenize different homogeneous catalysts on one solid support. The CAS route is the only viable strategy for the synthesis of polar bimodal polyethylene with good molecular level entanglement and minimal phase separation. This produces polyolefin materials with excellent mechanical properties, surface/dyeing properties, gas barrier properties, as well as extrudability and 3D-printability.

Currently, hundreds of different grades of polyolefins are commercially available with a wide variety of material properties[1]. This originates from an in-depth mechanistic understanding and thorough molecular control of the polymerization processes enabled by various transition-metal catalysts[2]. Controlling the polymerization process provides control over key parameters such as molecular weight and molecular weight distribution, which ultimately determine macroscopic material properties[3,4]. Molecular weight distribution is a critical parameter that determines many material properties[5]. Bimodal/multimodal polyethylene is an important specialty polyolefin that can be produced using a mixture of single-site catalysts (in solution or supported) in one reactor or by using one catalyst in a series of reactors under different conditions[6–9]. This type of material combines the superior properties of the low-molecular-weight fraction (stiffness, processability, etc.) and the high-molecular-weight fraction (mechanical strength, melt strength, etc.), making it capable of limiting the shear forces involved in extrusion[10,11].

The introduction of some polar functional groups into the otherwise non-polar backbone of polyolefins can improve many important properties[12–17]. The transition metal-catalyzed copolymerization of olefins with polar comonomers is the most direct route to access polar-functionalized polyolefins[18–29]. Due to the poisoning effect that polar groups have on the metal center, the incorporation of polar comonomers decreases the catalytic activity and polymer molecular weight. This inevitable trade-off makes this process uneconomical, and it is extremely difficult to access polar polyolefins with practical material properties. In this contribution, we tackle this issue through the targeted synthesis of bimodal polar-functionalized polyolefins (Fig. 1). In such a mixture, the high-molecular-weight fraction bears few functional groups and possesses high mechanical and melt properties. The low-molecular-weight fraction contained many functional groups, thereby implementing both polarity and processability.

Many factors such as molecular weight, molecular weight distribution, and branching content strongly influence the miscibility of polyolefin blends, which is crucial to realizing desired properties for multimodal polyolefins[30–35]. As expected, good miscibility is difficult to achieve for mixtures of linear and branched polyolefins[36,37]. This poor miscibility is exponentially amplified for the above-mentioned bimodal polar functionalized polyolefins, since the low-molecular-weight

[1]Hefei National Laboratory for Physical Sciences at the Microscale, CAS Key Laboratory of Soft Matter Chemistry, Department of Polymer Science and Engineering, University of Science and Technology of China, 230026 Hefei, China. [2]These authors contributed equally: Chen Zou, Quan Wang. ✉e-mail: siguifu@ustc.edu.cn; changle@ustc.edu.cn

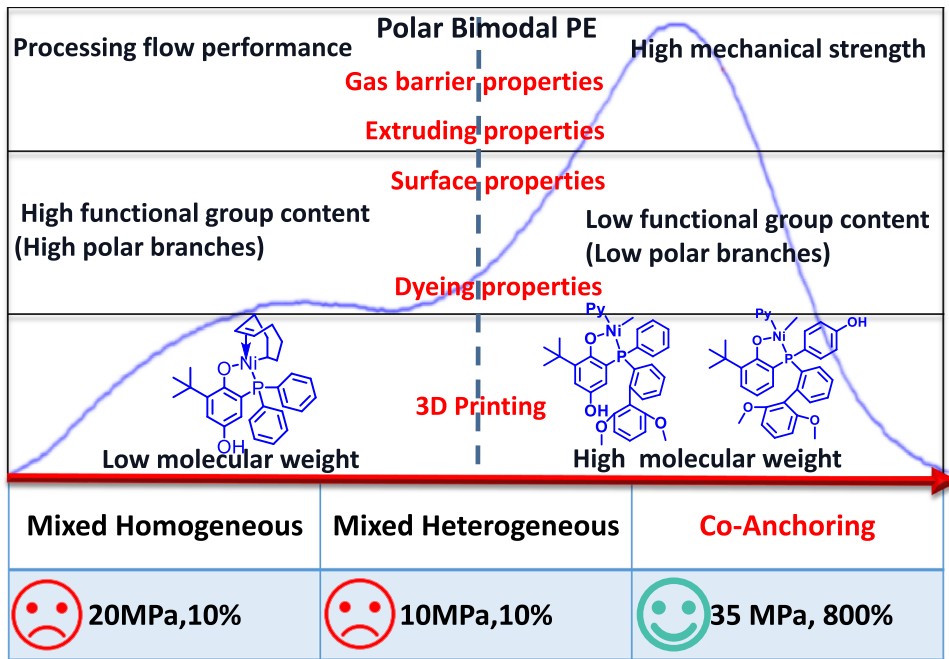

**Fig. 1 | Expected properties of polar bimodal polyethylene by combining low-molecular-weight and high-molecular-weight fractions.** Below are listed the tensile properties of polar bimodal polyethylene products prepared by mixed homogeneous, mixed heterogeneous, and co-anchoring strategies.

fraction is both more branched and more polar than the high-molecular-weight fraction.

In this contribution, in order to achieve molecular-level entanglement of the two distinct fractions, we explore three potential routes to access the target bimodal polar functionalized polyolefins: mixtures of homogeneous catalysts, mixtures of separately-supported heterogeneous catalysts, and a co-anchoring strategy (CAS) to heterogenize different homogeneous catalysts on one solid support.

## Results and discussion

### Heterogeneous catalysts

Previously, we developed an ionic anchoring strategy (IAS) for the heterogenization of transition metal catalysts through their interaction with solid supports with pre-installed ONa tags in the catalysts[38]. This strategy makes it very easy to co-anchor two or more homogeneous catalysts on one solid support to prepare bimodal polyethylene/polar polyethylene. In this work, sterically bulky **Ni1** and **Ni2** (prepared from the reaction of (Py)₂NiMe₂ precursor with phosphino-phenol ligands **L1** and **L2**) were selected to produce the high-molecular-weight fraction, and sterically-open **Ni3** was selected to produce the low-molecular-weight fraction. Our initial attempt to react ligand **L3** with (Py)₂NiMe₂ failed to give any isolable product. It may be due to the side reaction of this nickel precursor with the para-hydroxy group of ligand **L3** with small steric hindrance. Therefore, an alternative synthetic strategy was employed for the synthesis of **Ni3**. These homogenous nickel catalysts were mixed with NaH and MgO support to prepare heterogeneous catalysts **Ni1-MgO**, **Ni2-MgO**, and **Ni3-MgO**. In addition, mixtures of **Ni1/Ni2**, **Ni1/Ni3**, and **Ni2/Ni3** in different ratios were mixed with NaH and supported on MgO to prepare heterogeneous co-anchored catalysts **Ni1/Ni2-MgO**, **Ni1/Ni3-MgO**, and **Ni2/Ni3-MgO**. Mixtures of separately supported heterogeneous catalysts (**Ni1-MgO/Ni2-MgO**, **Ni1-MgO/Ni3-MgO**, and **Ni2-MgO/Ni3-MgO**) were also studied for comparison (Fig. 2).

### Bimodal polyethylene

Ethylene homopolymerization using these homogeneous and heterogeneous nickel catalysts was studied (Supplementary Table 1). Correlated with ligand sterics, the polyethylene molecular weight followed the order **Ni1** > **Ni2** > **Ni3** (Supplementary Table 1, entries 1–3). A similar order was observed for the supported nickel catalysts **Ni1-MgO** ($M_w$, 556.9 ×10⁴ g mol⁻¹) > **Ni2-MgO** ($M_w$, 177.9 × 10⁴ g mol⁻¹) > **Ni3-MgO** ($M_w$, 2.3 × 10⁴ g mol⁻¹) (Supplementary Table 1, entries 4–6). As expected, bimodal polyethylene was generated using co-anchored heterogeneous catalyst **Ni1/Ni2-MgO** (Supplementary Table 1, entries 7–10), along with tunable polymolecular weight distribution (4.9–9.6) with different catalyst ratios (1:2, 1:1, and 2:1). The utilization of **Ni1/N3-MgO** generated bimodal polyethylene composed of an ultra-high-molecular-weight fraction (millions) and low-molecular-weight fraction (tens of thousands), along with a polymolecular weight distribution in the range of 42.7–143.1 at different catalyst ratios (Supplementary Table 1, entries 11–15). The heterogeneous catalyst **Ni2/Ni3-MgO** generated bimodal polyethylene with a medium-molecular-weight fraction and low-molecular-weight fraction (Supplementary Table 1, entry 16). Mixed homogeneous catalyst **Ni1/Ni3** and mixed heterogeneous catalyst **Ni2-MgO/Ni3-MgO** also generated bimodal polyethylene (Supplementary Table 1, entries 17 and 18).

### Polar functionalized bimodal polyethylene

Subsequently, the copolymerization of ethylene with polar comonomers including *tert*-butyl acrylate (*t*BA) and methyl 10-undecenoate (UAE) was investigated (Table 1). For both comonomers, the comonomer incorporation ratio followed the order **Ni3-MgO** > **Ni1-MgO** > **Ni2-MgO**, while the copolymer molecular weight followed the reverse order (Table 1, entries 1–6). This was probably due to the steric effect induced by both the phosphine ligand and solid support. A higher copolymerization temperature led to a higher comonomer incorporation at the expense of copolymer molecular weight (Table 1, entry 7 vs entry 2).

Based on these results, polar functionalized bimodal polyethylene was prepared using co-anchored catalyst **Ni1/Ni3-MgO** (Table 1, entry 8; at **Ni1:Ni3** ratio of 1:1; $M_n$, 2.0 × 10⁴ g mol⁻¹, $M_w$, 23.1 × 10⁴ g mol⁻¹, PDI, 11.5). The co-anchored catalyst **Ni2/Ni3-MgO** led to the formation of copolymers with 0.7% comonomer incorporation and high molecular weight (Table 1, entry 9; $M_w$, 54.4 × 10⁴ g mol⁻¹). This was much higher than what was achievable using **Ni2-MgO** alone (Table 1, entry 7; $M_w$, 11.0 × 10⁴ g mol⁻¹). This co-anchoring strategy made it possible to

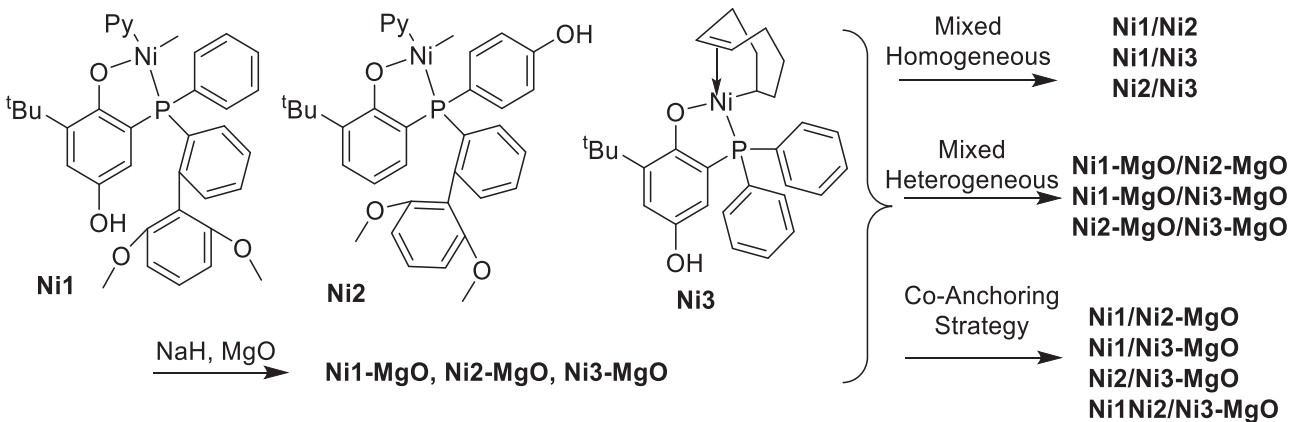

**Fig. 2 | Synthesis of desired catalysts.** Synthesis of mixed homogeneous catalysts, mixed heterogeneous catalysts, and co-supported catalysts by the co-anchoring strategy.

**Table 1 | Ethylene copolymerization with Ni catalysts[a]**

| Ent. | Cat. | Mon./mol/L | Yield/g[b] | Act.[b]/$10^5$ | $X_M$[c]/% | $T_m$[d]/°C | $M_w$[e]/$10^4$ | PDI[e] | $M_{w1}$[e]/$10^4$ | PDI$_1$[e] | $M_{w2}$[e]/$10^4$ | PDI$_2$[e] |
|---|---|---|---|---|---|---|---|---|---|---|---|---|
| 1 | **Ni1-MgO** | tBA/0.1 | 0.77 | 3.1 | 0.6 | 128.9 | 28.2 | 1.9 | | | | |
| 2 | **Ni2-MgO** | tBA/0.1 | 1.51 | 6.0 | 0.1 | 131.9 | 92.4 | 3.7 | | | | |
| 3 | **Ni3-MgO** | tBA/0.1 | 0.18 | 0.7 | 1.7 | 119.9 | 1.5 | 2.9 | | | | |
| 4 | **Ni1-MgO** | UAE/0.5 | 0.82 | 16.4 | 0.6 | 131.8 | 81.8 | 2.7 | | | | |
| 5 | **Ni2-MgO** | UAE/0.5 | 1.16 | 23.2 | 0.1 | 134.4 | 161.5 | 2.7 | | | | |
| 6 | **Ni3-MgO** | UAE/0.5 | 0.37 | 7.4 | 1.2 | 124.1 | 1.7 | 2.8 | | | | |
| 7[f] | **Ni2-MgO** | tBA/0.1 | 0.61 | 2.4 | 0.7 | 125.7 | 11.0 | 2.8 | | | | |
| 8 | **Ni1/Ni3-MgO (1:1)** | tBA/0.1 | 0.41 | 1.6 | 1.1 | 127.5 | 23.1 | 11.5 | 36.5 | 1.8 | 2.4 | 3.1 |
| 9 | **Ni2/Ni3-MgO (1:1)** | tBA/0.1 | 1.01 | 4.0 | 0.7 | 129.9 | 54.4 | 34.4 | 81.5 | 3.5 | 1.6 | 3.6 |
| 10[g] | **Ni1/Ni3-MgO (1:1)** | tBA/0.1 | 0.88 | 3.5 | 0.2 | 132.1 | 244.8 | 18.4 | 484.8 | 2.3 | 3.6 | 2.0 |
| 11 | **Ni1/Ni3-MgO (1:5)** | tBA/0.1 | 0.70 | 2.8 | 1.4 | 125.0 | 25.1 | 28.5 | 30.5 | 3.4 | 1.6 | 3.1 |
| 12 | **Ni2/Ni3-MgO(1:1)** | UAE/0.5 | 0.62 | 12.4 | 0.5 | 132.2 | 78.9 | 11.7 | 143.7 | 3.1 | 1.2 | 2.9 |
| 13 | **Ni2/Ni3-MgO (1:5)** | UAE/0.5 | 0.33 | 6.6 | 0.9 | 129.1 | 52.6 | 13.3 | 106.7 | 2.6 | 2.7 | 1.9 |
| 14[f] | **Ni2/Ni3-MgO (1:1)** | UAE/0.5 | 0.58 | 11.6 | 1.3 | 126.4 | 17.4 | 3.7 | 22.6 | 1.8 | 1.5 | 1.3 |
| 15[h] | **Ni2/Ni3-MgO (1:1)** | UAE/0.5 | 1.96 | 39.2 | 0.6 | 127.2 | 117.2 | 13.3 | 179.1 | 1.9 | 5.4 | 2.1 |
| 16 | **Ni1/Ni2/Ni3-MgO(1:1:1)** | UAE/0.5 | 0.81 | 16.2 | 0.7 | 130.8 | 93.4 | 53.9 | _[j] | _[j] | _[j] | _[j] |
| 17 | **Ni2/Ni3-TiO₂ (1:1)** | UAE/0.5 | 0.75 | 15.0 | 0.2 | 131.9 | 50.8 | 11.4 | 71.3 | 4.0 | 4.3 | 2.6 |
| 18 | **Ni2/Ni3-GF (1:1)** | UAE/0.5 | 1.21 | 24.2 | 0.2 | 131.0 | 92.0 | 11.8 | 112.1 | 4.4 | 5.6 | 2.6 |
| 19 | **Ni2/Ni3-APP (1:1)** | UAE/0.5 | 1.30 | 26.0 | 1.1 | 129.8 | 36.1 | 9.0 | 41.4 | 5.9 | 1.7 | 1.5 |
| 20 | **Ni2/Ni3-lignin (1:1)** | UAE/0.5 | 0.83 | 16.6 | 0.2 | 130.2 | 40.5 | 10.2 | 67.2 | 2.1 | 4.7 | 2.6 |
| 21[i] | **Ni2-MgO/Ni3-MgO** | tBA/0.1 | 1.01 | 4.0 | 0.6 | 118.7/133.9 | 56.0 | 34.2 | 79.7 | 4.0 | 1.3 | 3.4 |
| 22[i] | **Ni2-MgO/Ni3-MgO** | UAE/0.5 | 0.60 | 12.0 | 0.5 | 125.6/131.1 | 72.7 | 34.4 | 115.2 | 3.3 | 1.3 | 3.3 |

*GF* glass fiber, *APP* ammonium polyphosphate.
[a]Conditions: Entries 1–3, 7–11, and 21, cat. 5 μmol (Ni); Entries 4–6, 12–20, 22, cat. 1 μmol (Ni); 5 mL Heptane; $t = 30$ min; $T = 80$ °C; 8 atm.
[b]Yields are the average of at least two runs. Activity is in units of $10^5$ g/(mol cat. × h).
[c]Incorporation ratios of comonomers were determined from $^1$H NMR spectra.
[d]Determined by differential scanning calorimetry (DSC, second heating).
[e]$M_w$: $10^4$ g mol$^{-1}$, $M_n$, $M_w$, and PDI were determined by gel permeation chromatography in 1,2,4-trichlorobenzene at 160 °C. $M_{w1}$, PDI$_1$, $M_{w2}$, and PDI$_2$ were calculated by Gauss formula and fitting the copolymers generated by two nickel catalysts respectively. $M_n$, $M_{n1}$, and $M_{n2}$ are listed in Supplementary Table 2. The fitting curves are shown in Supplementary Fig. 1.
[f]$T = 120$ °C.
[g]$P = 30$ atm.
[h]$T = 120$ °C, $P = 30$ atm.
[i]Molar ratio of heterogeneous catalysts **Ni2-MgO** and **Ni3-MgO** was 1:1.
[j]The sample was generated by three kinds of nickel catalysts and cannot be accurately sealed and fitted.

prepare functional polyolefins with high comonomer incorporation while maintaining a high molecular weight. Furthermore, simply tuning the nickel catalyst ratios led to the formation of a series of polar bimodal polyolefins with different molecular weight fractions, comonomer incorporation, and molecular weight distribution values (Table 1, entries 10–15; PDI: 3.7–28.5). It is also possible to co-anchor

three homogeneous nickel catalysts (**Ni1**, **Ni2**, and **Ni3**) on the same solid support, allowing more versatility to tune the properties of the obtained polymers (Table 1, entry 16). Finally, the utilization of different functional solid supports (titanium oxide, glass fiber (GF), ammonium polyphosphate (APP), and lignin) generated functional bimodal polyethylene with different material properties such as

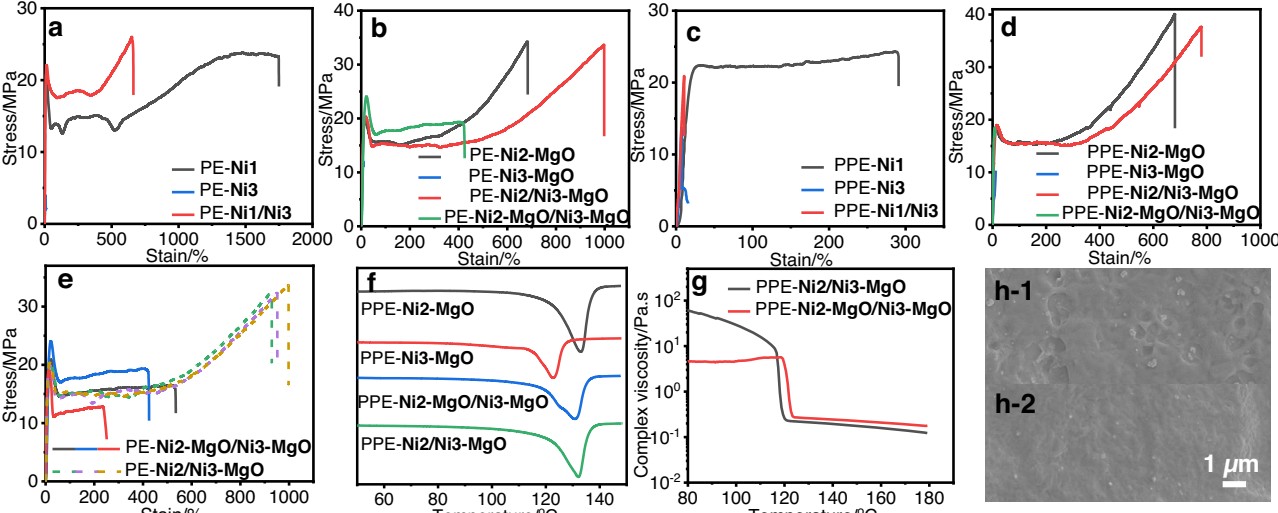

**Fig. 3 | Comparison of bimodal polyethylene prepared in different systems.**
**a** Tensile curves of polyethylene prepared by homogeneous polymerization. PE-**Ni1** (Supplementary Table 1, Entry 1), PE-**Ni3** (Supplementary Table 1, Entry 3), PE-**Ni1/Ni3** (Supplementary Table 1, Entry 17). **b** Tensile curves of polyethylene prepared by heterogeneous polymerization. PE-**Ni2-MgO** (Supplementary Table 1, Entry 5), PE-**Ni3-MgO** (Supplementary Table 1, Entry 6), PE-**Ni2/Ni3-MgO** (Supplementary Table 1, Entry 16), PE-**Ni2-MgO/Ni3-MgO** (Supplementary Table 1, Entry 18). **c** Tensile curves of polar polyethylene prepared by homogeneous polymerization. PPE-**Ni1** (Supplementary Table 3, Entry 1), PPE-**Ni3** (Supplementary Table 3, Entry 3), and PPE-**Ni1/Ni3** (Supplementary Table 3, Entry 4). **d** Tensile curves of polar

polyethylene prepared by heterogeneous polymerization. PPE-**Ni2-MgO** (Table 1, Entry 5), PPE-**Ni3-MgO** (Table 1, Entry 6), PPE-**Ni2/Ni3-MgO** (Table 1, Entry 12), and PPE-**Ni2-MgO/Ni3-MgO** (Table 1, Entry 22). **e** Repetitive tensile curves of polymers PE-**Ni2/Ni3-MgO** (Supplementary Table 1, Entry 16) and PE-**Ni2-MgO/Ni3-MgO** (Supplementary Table 1, Entry 18). **f** DSC curve of PPE-**Ni2-MgO** (Table 1, Entry 5), PPE-**Ni3-MgO** (Table 1, Entry 6), PPE-**Ni2-MgO/Ni3-MgO** (Table 1, Entry 22) and PPE-**Ni2/Ni3-MgO** (Table 1, Entry 12). **g** Rheological curve of PPE-**Ni2-MgO/Ni3-MgO** (Table 1, Entry 22) and PPE-**Ni2/Ni3-MgO** (Table 1, Entry 12). **h-1** SEM image of PPE-**Ni2-MgO/Ni3-MgO** (Table 1, Entry 22). **h-2** SEM image of PPE-**Ni2/Ni3-MgO** (Table 1, Entry 12). The original SEM images were listed in Supplementary Fig. 50.

photodegradability, flame retardancy, and oxidation resistance (Table 1, entries 17–20). Mixtures of the separately-supported heterogeneous catalyst **Ni2-MgO/Ni3-MgO** were also studied for reference (Table 1, entries 21–22).

## Mechanical properties of polar bimodal polyethylene

As expected, the mechanical properties of bimodal polyethylene (PE-**Ni1/Ni3**) prepared by mixed homogeneous catalysts **Ni1** and **Ni3** fell between those of PE-**Ni1** and PE-**Ni3** (Fig. 3a). This was also the case for the bimodal polyethylene (PE-**Ni2-MgO/Ni3-MgO**) prepared by separately mixing supported heterogeneous catalysts **Ni2-MgO** and **Ni3-MgO** versus those of PE-**Ni2-MgO** and PE-**Ni3-MgO** (Fig. 3b). This indicates good miscibility between the low-molecular-weight and high-molecular-weight fractions of polyethylene, regardless of whether they were generated using mixtures of homogeneous or heterogeneous nickel catalysts.

In direct contrast, the polar functionalized bimodal polyethylene (PPE-**Ni1/Ni3**) prepared using mixed homogeneous catalysts **Ni1** and **Ni3** showed very poor mechanical properties, despite the presence of the high-molecular-weight fraction generated by **Ni1** (Fig. 3c). Similarly, the polar bimodal polyethylene (PPE-**Ni2-MgO/Ni3-MgO**) prepared by using the mixed heterogeneous catalysts **Ni2-MgO** and **Ni3-MgO** showed very poor mechanical properties, despite the presence of high-molecular-weight fraction generated by **Ni2-MgO** (Fig. 3d). This was likely due to poor miscibility between the more-branched/more-polar low-molecular-weight fraction and the less-branched/less-polar high-molecular-weight fraction. The corresponding phase separation between these two fractions was detrimental to the mechanical properties.

Surprisingly, the bimodal polyethylene (PE-**Ni2/Ni3-MgO**) and polar bimodal polyethylene (PPE-**Ni2/Ni3-MgO**) prepared by using the co-anchored heterogeneous catalyst **Ni2/Ni3-MgO** showed comparable, even better, mechanical properties than those prepared by using **Ni2-MgO** alone (Fig. 3b, d). Clearly, the presence of the low-molecular-weight fraction in PE-**Ni2/Ni3-MgO** or PPE-**Ni2/Ni3-MgO** did not affect

the mechanical properties. It is hypothesized that the small distance between the two nickel centers in the co-anchored catalysts led to molecular-level entanglement of the low and high-molecular-weight fractions, thereby enabling good miscibility and even co-crystallization. Moreover, the reproducibility of the tensile tests of PE-**Ni2/Ni3-MgO** was much better than that of PE-**Ni2-MgO/Ni3-MgO**, indicating a miscibility issue and phase separation in the latter case (Fig. 3e).

Differential scanning calorimetry (DSC) showed the presence of two melting points for PPE-**Ni2-MgO/Ni3-MgO**, which corresponded to PPE-**Ni2-MgO** and PPE-**Ni3-MgO**, respectively (Fig. 3f). In contrast, only one melting point was observed for PPE-**Ni2/Ni3-MgO** (Fig. 3f). This was observed for all polymers generated using co-anchored catalysts (Supplementary Figs. 2–41). However, the bimodal polyethylene prepared by using a mixed homogeneous catalyst or heterogeneous catalysts had two melting points (Supplementary Figs. 42–49). And the complex viscosities of PPE-**Ni2/Ni3-MgO** and PPE-**Ni2-MgO/Ni3-MgO** were determined by temperature-sweep experiments at a scan rate of 1 °C min$^{-1}$, and a frequency of 1.0 Hz (Fig. 3g), the polar bimodal polyethylene PPE-**Ni2/Ni3-MgO** shown higher viscosity, indicating more entangled molecular chains. In addition, the SEM image of polar bimodal polyethylene PPE-**Ni2-MgO/Ni3-MgO** indicated the presence of more phase separation than PPE-**Ni2/Ni3-MgO** (Fig. 3h-1 vs Fig. 3h-2). These results are consistent with the above-mentioned mechanical studies and further support the hypothesis concerning the molecular weight entanglement and co-crystallization of the low and high-molecular-weight fractions using the co-anchoring strategy.

We have prepared a series of bimodal polyethylene samples with different comonomer incorporation ratios using co-anchored catalyst and mixed heterogeneous catalyst (Supplementary Table 4), and compared their phase compatibility, mechanical properties and rheological properties. At low comonomer incorporation ratio (<0.5%), SEM images showed uniform homogeneity for both cases (Fig. 4). However, the samples prepared by mixed heterogeneous catalyst **Ni2-MgO/Ni3-MgO** showed obvious phase separation at incorporation

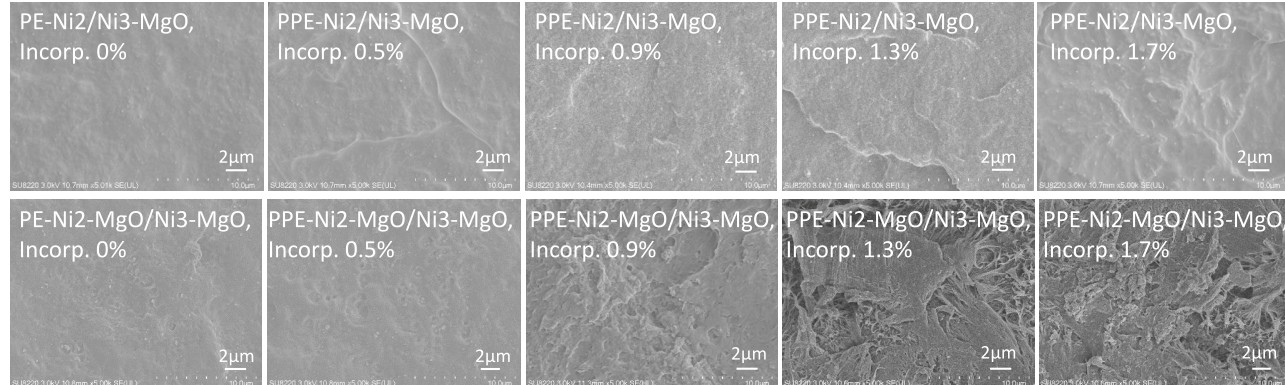

**Fig. 4 | Comparison of SEM images of polar bimodal polyethylene samples prepared by co-anchored catalyst and mixed heterogeneous catalyst after incorporation of polar monomer.** These bimodal polymers in the first row were prepared by co-anchored catalyst **Ni2/Ni3-MgO**, and those in the second row were prepared by mixed heterogeneous catalyst **Ni2-MgO/Ni3-MgO**. Incorp. (Incorporation) ratios of comonomers were determined from ¹H NMR spectra. The characterization data of these polymers are listed in Supplementary Information Table 4.

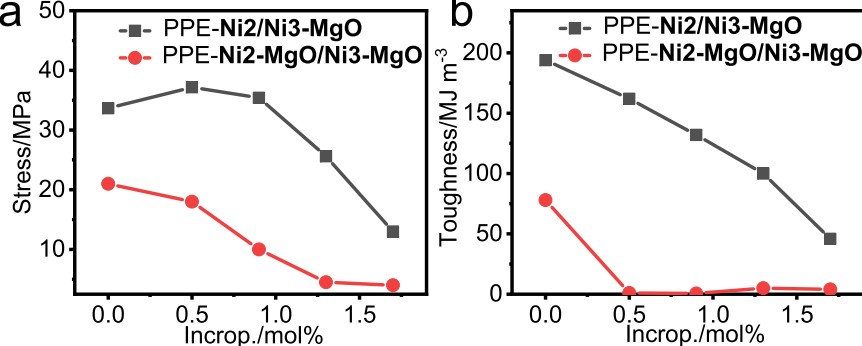

**Fig. 5 | Correlation diagram of mechanical properties with polar monomer incorporation of a series of bimodal polyethylene samples. a** Correlation diagram of tensile strength with polar monomer incorporation. **b** Correlation diagram of toughness with polar monomer incorporation. The characterization data of these polymers are listed in Supplementary Information Table 4.

ratios of above 0.9%. In direct contrast, the samples prepared using co-anchored catalyst **Ni2/Ni3-MgO** maintained great compatibility even at high comonomer incorporation (1.7%).

Similar with the SEM results, the mechanical properties of the samples prepared by co-anchoring strategy were only slightly decreased with increasing comonomer incorporation (0–1.7%) (Fig. 5 and Supplementary Fig. 51). However, the samples prepared by mixed heterogeneous catalyst showed extremely poor mechanical properties at comonomer incorporation ratio of above 0.5%, due to the obvious phase separation of the two components. In particular, the toughness of the material decreases sharply and almost disappears after the introduction of polar monomer. Clearly, the mechanical properties of bimodal polymer samples prepared by the two catalyst systems are quite different due to the differences of their microscopic phase separation behaviors.

The comparison of rheological properties of these samples also showed that the complex viscosity of bimodal polyethylene prepared by co-anchoring strategy were much higher than those prepared by mixed heterogeneous catalyst before the melting temperature, indicating that the two components of bimodal polyethylene prepared by co-anchoring strategy are more entangled (Supplementary Fig. 51). In addition, similar results were observed for other types of supported heterogeneous catalysts (APP) (Supplementary Fig. 52).

### Polar content and surface properties

The co-anchoring strategy enabled the formation of polar polyolefin materials with both high polar comonomer incorporation and excellent mechanical properties, which is very difficult to access using traditional methods. For example, **Ni2-MgO** afforded a polar polyolefin with outstanding mechanical properties (Table 1, Entry 2; stress: 36.5 MPa, strain: 880%) with only 0.1% comonomer incorporation (Fig. 6a). Increasing the comonomer incorporation to 0.7% (Table 1, entry 7) significantly decreased the molecular weight and mechanical properties (stress: 22.0 MPa, strain: 50%). In contrast, the co-anchored catalyst **Ni2/Ni3-MgO** generated polar bimodal polyethylene with the same level of comonomer incorporation along with great mechanical properties (Table 1, Entry 9; incorporation: 0.7%, stress: 30.1 MPa, strain: 880%).

The incorporation of polar comonomers is expected to modify the surface properties of polyolefin materials. The surface properties of selected polar bimodal polyethylenes (Fig. 6b and Supplementary Fig. 54, Table 1, entries 8–11, incorporation, 0.2–1.4%) were studied by measuring their water contact angles (WCAs)[39]. Generally, a higher comonomer content led to a lower WCA. Interestingly, the polar bimodal polyethylene prepared by the co-anchoring strategy showed a significantly lower WCA than the unimodal polar polyethylene at the same comonomer content. The highly polar low-molecular-weight fraction may have tended to aggregate on the surface of the measured sample.

### Dyeing properties

The dyeing properties of polyolefins are related to their surface properties[40,41]. The polar bimodal polyethylene and dye powder (2,2′-[(3,3′-dichloro[1,1′-biphenyl]-4,4′-diyl)bis(azo)]bis[*N*-(2-methylphenyl)-3-oxobutyramide]) were mixed. The blends were melt-pressed at

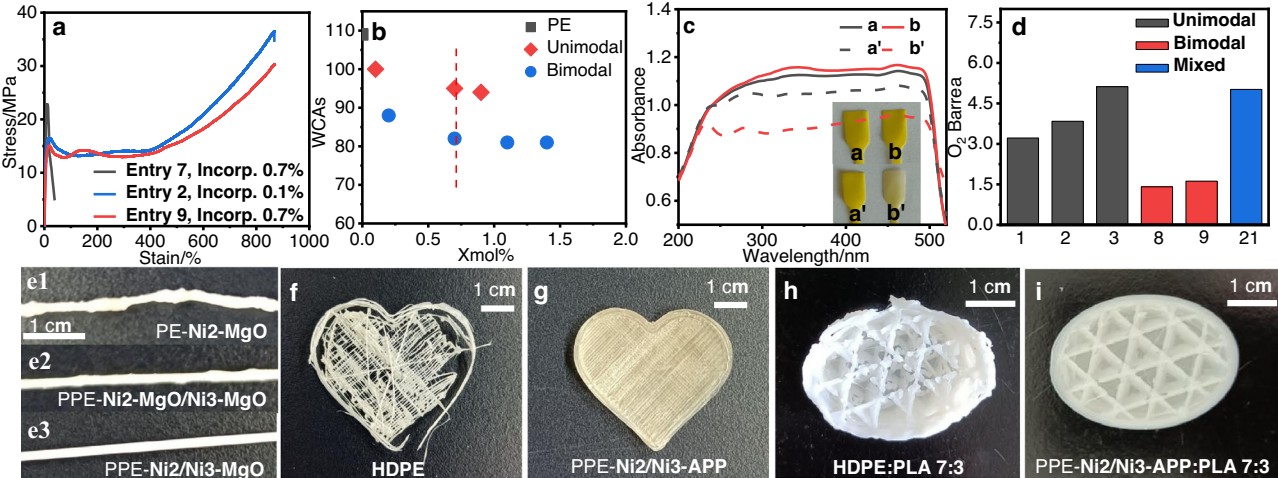

**Fig. 6 | Polar properties and 3D printing of polar bimodal polyethylene.**
**a** Tensile curves of samples from Table 1, Entry 7 (Incorp. 0.7%), Entry 2 (Incorp. 0.1%), and Entry 9 (Incorp. 0.7%). **b** Water contact angles of ethylene/*tert*-butyl acrylate copolymers from Table 1. Unimodal: unimodal polar polyethylene from Table 1, Entry 1, Entry 2, and Entry 7. Bimodal: polar bimodal polyethylene prepared by co-anchored catalyst from Table 1, Entries 8–11. **c** UV-vis absorption spectra of the dyed polymer products before and after acetone wash (a and a': Table 1, Entry 9. b and b': Table 1, Entry 7). **d** Oxygen permeability coefficient of polyethylene samples from Table 1 at 25 °C. The abscissa represents the relative entry in Table 1.

Unimodal: unimodal polar polyethylene from Table 1. Bimodal: polar bimodal polyethylene prepared by co-anchored catalyst, Mixed: prepared by mixed heterogeneous catalyst. **e** Images of extruded samples of PE-**Ni2-MgO** (Supplementary Table 1, Entry 5), PPE-**Ni2-MgO/Ni3-MgO** (Table 1, Entry 22) and PPE-**Ni2/Ni3-MgO** (Table 1, Entry 12). The enlarged images were listed in Supplementary Fig. 53. **f–i** 3D-printed samples of commercial HDPE, PPE-**Ni2/Ni3-APP** (Table 1, Entry 19), HDPE: PLA 7:3 (prepared by blending commercial HDPE and polylactic acid in a ratio of 7 to 3) and PPE-**Ni2/Ni3-APP**: PLA 7:3 (prepared by blending PPE-**Ni2/Ni3-APP** and polylactic acid in a ratio of 7 to 3).

150 °C to obtain test specimens (Fig. 6c). Subsequently, the specimens were washed with hot acetone for 72 h and dried in a vacuum oven to a constant weight. In the UV-vis absorption spectra, the absorbance peak at 465 nm for polar bimodal polyethylene only decreased slightly after washing (Table 1, Entry 9). In contrast, the unimodal polar polyethylene with the same comonomer content (Table 1, Entry 7) showed a much more dramatic decrease. This is consistent with the surface property studies using WCAs and indicates the superior surface properties of the polar bimodal polyethylene prepared using the co-anchoring strategy, which further indicates that the surface of the film prepared by melting processing may contain more polar functional groups.

## Gas barrier properties

The gas barrier properties of polymer films are very important for applications such as food packaging. After melting and pressing polar bimodal polyethylene and unimodal polyethylene prepared by different catalyst systems into thin films, we use GAS PERMEABILITY TESTER to measure the barrier of the film to oxygen. The results showed that the oxygen barrier properties of polar bimodal polyethylene prepared by co-anchored catalysts were even better than those of the unimodal polymer sample (Fig. 6d; 3.22 vs 1.41, 3.84 vs 1.62). It is possible that the low-molecular-weight fraction, especially its good miscibility and entanglement with the high-molecular-weight fraction, made the film denser and improved its oxygen barrier properties. Furthermore, the bimodal copolymer prepared by mixed heterogeneous catalyst **Ni2-MgO/Ni3-MgO** showed poor oxygen barrier performance (5.02). This may be due to great phase compatibility of the polar bimodal polyethylene prepared by co-anchored catalyst, leading to high molecular chain entanglement and the formation of dense film.

## Extruding properties

Bimodal polyethylene can improve processability[7]. The extrusion properties of polar bimodal polyethylene samples were studied with a single-screw extruder at 200 °C (Fig. 6e). Clearly, the extruded polar bimodal polyethylene sample (e2 and e3) was much smoother and more uniform than the sample with non-polar polyethylene

(e1, Supplementary Table 1, entry 5, PE-**Ni2-MgO**). The introduction of polar functional groups and the presence of low-molecular-weight copolymers may have both contributed to its good extrusion performance. The extrusion performance of polar bimodal polyethylene prepared by co-anchoring strategy (e3, Table 1, Entry 12, PPE-**Ni2/Ni3-MgO**) was better than that from separately mixed heterogeneous catalyst (e2, Table 1, Entry 22, PPE-**Ni2-MgO/Ni3-MgO**), which further indicated that the two components of polar bimodal polyethylene prepared by co-anchoring strategy had better blending properties.

## 3D printing

The emergence of three-dimensional (3D) printing has added a new dimension to polymer processing and holds huge prospects for manufacturing complex multi-functional material systems in a single processing step[42,43]. However, 3D printing high-density polyethylene (HDPE) has been problematic owing to its massive shrinkage, accompanied by its poor adhesion to common build plates[44]. Thus, it is difficult to 3D print commercial HDPE (Fig. 6f). The polar bimodal polyethylene material enabled by the co-anchoring strategy showed both improved extrusion properties and good surface properties. These properties made 3D printing viable for these polar polyethylene materials (Fig. 6g). The deliberately-chosen APP support also rendered the material flame retardant properties (Table 1, entry 19). The good surface properties of these polar bimodal polyethylene samples also enabled good compatibility with other types of polymers, making them more versatile for tuning material properties. For example, as shown in Supplementary Fig. 55, after blending non-polar HDPE with polylactic acid (PLA), the SEM image showed obvious "sea-island" phase separation. However, the polar bimodal polyethylene PPE-**Ni1/Ni3-APP** with polar groups showed excellent polar compatibility, therefore, it was much easier to 3D print blends of PLA with polar bimodal polyethylene versus commercial HDPE (Fig. 6h vs Fig. 6i).

Here, a co-anchoring strategy (CAS) was developed to heterogenize different homogeneous catalysts on one solid support. This strategy led to the formation of polar bimodal polyethylene, with a low-polarity linear high-molecular-weight faction and high-polarity branched low-molecular-weight fraction. Mechanical properties, DSC,

and SEM studies indicated good miscibility and minimal phase separation between the two fractions. Moreover, the polar bimodal polyethylene prepared by this co-anchoring strategy had excellent surface properties, dyeing properties, extrusion properties, and could be 3D printed. The utilization of different solid supports can potentially induce different functionalities. It is expected that this strategy will inspire more practical applications for polar functionalized polyolefin materials.

## Methods
### General methods and materials
All experiments were carried out under a dry nitrogen atmosphere using standard Schlenk techniques or in a glovebox. Dichloromethane, THF, and heptane were purified in solvent purification systems. Deuterated solvents used for Liquid NMR spectroscopy were dried and distilled prior to use. Liquid NMR spectra were recorded on a JNM-ECZR/S1 spectrometer at ambient temperature unless otherwise stated. The chemical shifts of the $^1$H and $^{13}$C NMR spectra were referenced to tetramethylsilane. Coupling constants are in Hz. Molecular weight and molecular weight distribution of the polymer were determined by gel permeation chromatography (GPC) with a PL-220 equipped with two Agilent PLgel Olexis columns at 160° C using 1,2,4-trichlorobenzene as a solvent, and the calibration was made using polystyrene standard and are corrected for linear polyethylene by universal calibration using the Mark–Houwink parameters of Rudin: $K = 1.75 \times 10^{-2}$ cm$^3$ g$^{-1}$ and $R = 0.67$ for polystyrene and $K = 5.90 \times 10^{-2}$ cm$^3$ g$^{-1}$ and $R = 0.69$ for polyethylene. DSC measurements were performed on a TA Instruments DSC 250. Samples (ca. 5 mg) were annealed by heating to 150 °C at 10 °C min$^{-1}$, cooled to 40 °C at 10 °C min$^{-1}$, and then analyzed while being heated to 150 °C at 10 °C min$^{-1}$. Powders of TiO$_2$ (20–40 nm), MgO (-20 nm) were obtained from Nanjing XFNANO Materials Tech Co., Ltd., China. These Powders were treated in a tube furnace at 600 °C for 6 h before use. APP ($n > 1000$) was purchased from Energy Chemistry. Glass fiber (8000 mesh) was purchased from Fuhua Nano New Materials Company. Glass fiber, lignin and APP were all treated with a certain amount of MAO (20 times equivalent of catalyst) before use.

### Mechanical properties
Stress/strain experiments were performed at room temperature at 10 mm min$^{-1}$ using a universal testing machine, Suns UTM 2502 from Shenzhen Suns Technology Company. At least three specimens of each polymer were tested. Each polymer was melt-pressed at about 150 °C, 5 MPa for 5 min to obtain the test specimen. The dumbbell shaped test specimens had the following dimensions: gauge length, 28 mm; width, 2 mm; and thickness, about 1 mm.

### Rheological experiment
Complex viscosity was determined by temperature-sweep experiments using an Anton Paar MCR302 rheometer (plate: 25 mm diameter). The temperature scan rate was set to 1 °C min$^{-1}$, and the frequency was set to 1.0 Hz.

### Scanning electron microscopy (SEM)
The images of fracture surface for polymers and energy dispersive spectroscopy (EDS) were obtained using a Hitachi Model X650 SEM system. The polymer was melt-pressed at about 150 °C, 5 MPa for 5 min to obtain the test specimen, and freeze it with liquid nitrogen and then wetting-off to obtain the fracture surface for SEM observation.

### Dyeing experiment
A mixture of 20 mg of dye powder (2,2′-[(3,3′-Dichloro[1,1′-biphenyl]-4,4′-diyl)bis(azo)]bis[N-(2-methylphenyl)-3-oxobutyramide]) and 1 g of copolymer was stirred in 50 mL of Tol. at 100 °C for 100 min. After the solvent was drained, the polymer sample was pressed to form a film at

150 °C, 5 MPa. Subsequently, the specimens were washed with hot acetone for 72 h and dried in a vacuum oven to a constant weight. Used the UV-vis absorption analysis after drying by SolidSpec-3700DUV.

### Gas barrier experiment
Oxygen permeability was determined by GAS PERMEABILITY TESTER (Basic 201, Jinan Languang Electromechanical Technology Co., Ltd) according to GB/T 1038-2000 at 25 °C. The polymer was melted and pressed at about 150 °C, 5 MPa for 10 min to obtain a film with the thickness of about 0.5 mm, and cut it into a circle with the diameter of 10 mm for gas barrier experiment. Install the prepared film into the vacuum chamber of the gas permeability tester, inject oxygen, measure the permeability of the film after a certain time, and prepare at least three samples of each polymer for testing.

### Water contact angle measurement
Water contact angles on polymer films were measured with Contact Angle Meter SL200B (Solon Tech. Co., Ltd.) by the dynamic sessile drop method. The polymer was melted and pressed at about 150 °C, 5 MPa for 5 min to obtain a polymer film for testing. The water contact angles of the polymer thin films were measured using a contact angle goniometer at 25 °C with an accuracy of ±3°. The reported values are the average of at least three measurements made at different positions of the film.

### Extruding experiment
The extrusion properties of these polyethylene were studied with a single screw extruder. Extrusion temperature, 200 °C, die diameter, 1.75 mm.

### 3D printing
The consumables for 3D printing experiment were pre extruded at 200 °C by a single screw extruder, die diameter, 1.75 mm. The 3D Printing experiment was completed by Reality CR-3040 Pro (Creality) Parameter: Nozzle diameter: 0.8 mm, Nozzle temp.: 200 °C, Bed temp.: 30 °C, Printing speed: 5 mm s$^{-1}$, Layer height: 0.2 mm, Cooling of the printed object: disabled.

### Procedure for polymerization
In a typical experiment, a Biotage Endeavor Parallel Pressure Reactor with 8 built-in parallel high-pressure polymerization reactor each with a volume of 10 mL was used for ethylene polymerization. After adding a certain amount of catalyst and 5 mL solvent at desired temperature, ethylene was inputted to start polymerization. At the end of the polymerization, the polymer product was filtered and dried at 45 °C for 24 h under vacuum. The copolymer product was filtered, extracted with a Soxhlet extractor to remove remaining comonomer, and dried at 45 °C for 24 h under vacuum.

## Data availability
All data necessary to support the conclusions of this paper are available in the Supplementary Information, including materials, detailed experimental procedures, and characterization, as well as DSC (Supplementary Figs. 2–49), Water contact angle (Supplementary Fig. 54), Original data of stress–strain curve (Supplementary Fig. 56), SEM (Supplementary Figs. 52, 55, and 57), NMR data (Supplementary Figs. 58–95), GPC (Supplementary Figs. 96–143). All data can be requested from the authors upon request.

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

## Acknowledgements
This work was supported by the National Key R&D Program of China (No. 2021YFA1501700), National Natural Science Foundation of China (Nos. 52025031, 52203016, U19B6001 and U1904212), and China Postdoctoral Science Foundation (2021M703072 and 2022T150617).

## Author contributions
C.C. and G.S. conceived the project. C.Z. and Q.W. performed experiments regarding the synthesis and characterization. All authors contribute to data analysis and paper writing.

## Competing interests
The authors declare no competing interests.
