## [Peer Review File · Nature Communications]

A Co-Anchoring Strategy for the Synthesis of Polar Bimodal PolyethyleneReviewers' Comments:

Reviewer #1:

Remarks to the Author:

In this paper mixtures of catalysts are co-anchored onto a MgO (or other) support using strong electrostatic adsorption previously developed by this group. I do not find this to be a weakness because the article focuses almost exclusively on the properties of the polymers as opposed to methodological development of catalyst properties. The behavior observed by the authors is not entirely unique, similar bimodal polymerization behavior was observed recently in a single site heterogeneous Pd catalyst catalyst that also incorporates polar monomers (ref 19), though that study's primary focus was catalyst development and detailed properties of the polymers were not discussed extensively. I do feel this study could be of interest to readers of Nature Communications, provided the following comments are addressed by the authors.

Table 1 treats all polymers as a single MW distribution, but this is incorrect. The table should be revised to reflect the different MW distributions produced by these catalysts. For example, the dispersity of entry 9 is certainly not 34.4, but certainly two dispersity values closer to what is observed for the two co-supported catalysts.

I am somewhat confused by Figure 1 h-1/h-2. These look like solution processed polymers, but this is not clearly stated. Also, the phase separation argument is not particularly strong here. I am surprised that the SEM in both examples are so smooth, it almost looks like nothing is there, which is surprising and not totally expected. Measuring these samples at high magnification is needed to support the arguments.

Somewhat related, the 3D printing data is compelling, and I wonder if SEM data for products in Figure 2H and 2I would be a more compelling proof-of-concept experiment showing how phase segregation does (or does not) occur with the PPE materials.

Reporting images of extrudate of PPE with mixtures of Ni₂/MgO and Ni₃/MgO should be given. Also, the images are not particularly clear, it was not obvious to me what I should be looking at in these images.

A stylistic point, but one the authors should consider revising: many of the properties discussed on page 5 (dyeing, O₂ permeability, and 3D printing) are written like a progress report. In addition, more description of how the gas barrier properties are measured would be helpful to the general audience.

Several minor typos in the need to be corrected (D6H6, O₂ barrea, etc)

Reviewer #2:

Remarks to the Author:

The manuscript "A Co-Anchoring Strategy for the Synthesis of Polar Bimodal Polyethylene" describes the co-anchoring of Ni olefin polymerization catalysts to MgO for the purposes of making blends of high MW HDPE (made by catalyst 1 or 2) with lower Mw functionalized PE (made by catalyst 3). The performance of the supported catalysts is better than the homogeneous analogs, and co-anchored mixtures is better than mixtures of the individual heterogeneous catalysts. The tensile properties, rheology, melting behavior, wetting studies, oxygen permeability, and 3D printing results were reported and support the hypothesis that the polar functionality enhances the material properties of the PE blends.

I have some questions about catalyst 3, however. The authors report that a complex with the formula (L)Ni(COD) is formed where L is the mono-deprotonated ligand; however, not enough information is provided. As written, the complex is formally Ni(I) and would be paramagnetic, but NMR data is

reported that is not consistent with this hypothesis. I would expect that either a Ni(0) complex (LHNiCOD) or Ni(II) complex (LNiCODH) would form, where CODH is bound to Ni as an alkyl or allylic substituent. I have drawn some of these in the attached file. The NMR data are listed but not assigned, and the spectrum is not shown in the SI. Likewise the elemental analysis (both calculated and experimentally observed) was listed for the LNi fragment (C₂₂H₂₂NiO₂P) rather than LNi(COD) (C₃₀H₃₅NiO₂P). It is also unclear why Ni(COD)₂ was used as a precursor rather than (py)₂NiMe₂, which was used for complexes 1 and 2, or other Ni(II) precursors such as Ni(allyl)Br dimer, py₂Ni(CH₂TMS)₂, (PR₃)₂NiPhCl, etc.

Although the syntheses of the catalysts is not really the focus of the paper, the lack of characterization for catalyst 3, or even a discussion of previous studies are concerning given that catalyst 3 is the one making the functionalized polymer. For this reason, I cannot recommend this manuscript for publication. I will be happy to re-review it once these questions are addressed.

complex 3 (COD binding mode not described)

Empirical formula used for elemental analysis

Ni(I), paramagnetic?

Chemical Formula: $C_{22}H_{22}NiO_2P$

Molecular Weight: 408.08

Elemental Analysis: C, 64.75; H, 5.43; Ni, 14.38; O, 7.84; P, 7.59

or

or

Chemical Formula: $C_{30}H_{35}NiO_2P$

Molecular Weight: 517.28

Elemental Analysis: C, 69.66; H, 6.82; Ni, 11.35; O, 6.19; P, 5.99

Reviewer #3:

Remarks to the Author:

Zou and coworkers report a strategy for the synthesis of blends of two polymers: 1) a low MW fraction with high levels of polarity, and 2) a higher MW fraction with fewer polar groups. It is proposed that such a mixture will have the ideal properties of good mechanical properties, along with beneficial properties (dyeing, gas barrier, extrusion printing, etc). To achieve miscibility between these two components, three synthetic routes were explored: mixtures of homogeneous catalysts, separately supported heterogeneous catalysts, and a co-anchoring strategy. It was claimed that the co-anchoring strategy worked better than the other two strategies.

First, I feel that this paper addresses an important topic, and that the science here is excellent. However, I believe that the paper is fairly applied, and will likely be of interest to a select group of scientists working in the area of functional polyolefins, rather than a broad scientific audience. For these reasons, it is my opinion that this work would be better suited to a more specialized journal focusing on polymer synthesis. I would consider this for Nature Communications if the work were less empirical. For example, it is unclear to me how different the levels of functionality can be while still achieving miscibility. I would assume that at some gap in functionality, that the materials would phase separate. If the authors could make an array of PE materials with varying levels of functional group incorporation using single catalysts under controlled conditions, then map out the phase space for miscibility or phase separation, I would view this to significantly improve the scientific component of the paper. As it stands, I feel it would be better suited for a more specialized polymer journal.

As an added note, I believe Ref 34 is incorrect.

Reviewer #1 (Remarks to the Author):

In this paper mixtures of catalysts are co-anchored onto a MgO (or other) support using strong electrostatic adsorption previously developed by this group. I do not find this to be a weakness because the article focuses almost exclusively on the properties of the polymers as opposed to methodological development of catalyst properties. The behavior observed by the authors is not entirely unique, similar bimodal polymerization behavior was observed recently in a single site heterogeneous Pd catalyst catalyst that also incorporates polar monomers (ref 19), though that study's primary focus was catalyst development and detailed properties of the polymers were not discussed extensively. I do feel this study could be of interest to readers of Nature Communications, provided the following comments are addressed by the authors.

Table 1 treats all polymers as a single MW distribution, but this is incorrect. The table should be revised to reflect the different MW distributions produced by these catalysts. For example, the dispersity of entry 9 is certainly not 34.4, but certainly two dispersity values closer to what is observed for the two co-supported catalysts.

Answer: Thanks a lot for your comments. According to the reviewer's comments, we have processed part of the data in the supporting information. These Peak fitting data were calculated by Gauss formula.

Table S2. Peak fitting data of molecular weight of polar bimodal copolymers in Table 1.^a

Ent.	Cat.	Mon. (mol/L)	M_n^e (10^4)	M_w $^e(10^4)$	M_w/M_n^e	M_{n1} $^e(10^4)$	M_{w1} $^e(10^4)$	M_{w1}/M_{n1}^e	M_{w2} $^e(10^4)$	M_{n2} $^e(10^4)$	M_{w2}/M_{n2}^e
1	Ni1-MgO	t BA(0.1)	14.6	28.2	1.9						
2	Ni2-MgO	t BA(0.1)	24.7	92.4	3.7						
3	Ni3-MgO	t BA(0.1)	0.5	1.5	2.9						
4	Ni1-MgO	UAE(0.5)	30.7	81.8	2.7						
5	Ni2-MgO	UAE(0.5)	58.9	161.5	2.7						
6	Ni3-MgO	UAE(0.5)	0.6	1.7	2.8						
7 ^b	Ni2-MgO	t BA(0.1)	3.9	11.0	2.8						
8	Ni1/Ni3-MgO(1:1)	t BA(0.1)	2.0	23.1	11.5	36.5	20.7	1.8	2.4	0.8	3.1
9	Ni2/Ni3-MgO(1:1)	t BA(0.1)	1.6	54.4	34.4	81.5	23.3	3.5	1.6	0.4	3.6
10 ^b	Ni1/Ni3-MgO(1:1)	t BA(0.1)	13.3	244.8	18.4	484.8	213.6	2.3	3.6	1.7	2.0
11	Ni1/Ni3-MgO(1:5)	t BA(0.1)	0.9	25.1	28.5	30.5	9.0	3.4	1.6	0.5	3.1
12	Ni2/Ni3-MgO(1:1)	UAE(0.5)	6.8	78.9	11.7	143.7	46.2	3.1	1.2	0.4	2.9
13	Ni2/Ni3-MgO(1:5)	UAE(0.5)	4.0	52.6	13.3	106.7	40.6	2.6	2.7	1.4	1.9
14 ^c	Ni2/Ni3-MgO(1:1)	UAE(0.5)	4.7	17.4	3.7	22.6	12.3	1.8	1.5	1.2	1.3
15 ^d	Ni2/Ni3-MgO(1:1)	UAE(0.5)	8.8	117.2	13.3	179.1	95.7	1.9	5.4	2.6	2.1

16	Ni1/Ni2/Ni3-MgO(1:1:1)	UAE(0.5)	1.7	93.4	53.9	_f	_f	_f	_f	_f	_f
17	Ni2/Ni3-TiO2(1:1)	UAE(0.5)	4.5	50.1	11.4	71.3	17.9	4.0	4.3	1.7	2.6
18	Ni2/Ni3-GF(1:1)	UAE(0.5)	7.8	92.0	11.8	67.2	32.3	2.1	4.7	1.8	2.6
19	Ni2/Ni3-APP(1:1)	UAE(0.5)	4.0	36.1	9.0	41.4	7.1	5.9	1.7	1.1	1.5
20	Ni2/Ni3-lignin(1:1)	UAE(0.5)	4.0	40.5	10.2	67.2	32.3	2.1	4.7	1.8	2.6
21 ^e	Ni2-MgO/Ni3-MgO	tBA(0.5)	1.6	56.0	34.2	79.7	20.1	4.0	1.3	0.4	3.4
22 ^e	Ni2-MgO/Ni3-MgO	UAE(0.5)	2.1	72.7	32.4	115.2	35.4	3.3	1.3	0.4	3.3

^a Conditions: 1-3, 7-11, and 21, cat. 5 μmol (Ni); 4-6, 12-20, 22, cat. 1 μmol (Ni); 5 mL Hex.; $t = 30$ min; $T = 80$ $^{\circ}\text{C}$; 8 atm. ^c M_n : 10^4 g mol⁻¹, M_n , M_w , and M_w/M_n were determined by gel permeation chromatography in 1,2,4-trichlorobenzene at 160 $^{\circ}\text{C}$. M_{n1} , M_{w1} , M_{w1}/M_{n1} , M_{n2} , M_{w2} , and M_{w2}/M_{n2} were calculated by Gauss formula and fitting the copolymers generated by two nickel catalysts respectively. GF: glass fiber; APP: ammonium polyphosphate. ^b $T = 120$ $^{\circ}\text{C}$, 8 atm. ^c $T = 80$ $^{\circ}\text{C}$, 30 atm. ^d $T = 120$ $^{\circ}\text{C}$, 30 atm. ^e Molar ratio of heterogeneous catalysts Ni2-MgO and Ni3-MgO was 1: 1. ^f The sample was generated by three kinds of nickel catalysts and cannot be accurately sealed and fitted.

Gauss formula

$$y = y_0 + \frac{A}{w\sqrt{\pi/2}} e^{-2\frac{(x-x_c)^2}{w^2}}$$

$A > 0$
 offset: $y_0 = 0$
 center: $x_c = 0$
 width: $w = 2$
 area: $A = 1$
 $y_c = y_0 + A / (w \cdot \sqrt{\pi/2})$
 $w = \text{FWHM} / \sqrt{\ln(4)}$

Fig. S1. Peak fitting curves of molecular weight of polar bimodal copolymers in Table 1.

I am somewhat confused by Figure 1 h-1/h-2. These look like solution processed polymers, but this is not clearly stated. Also, the phase separation argument is not particularly strong here. I

am surprised that the SEM in both examples are so smooth, it almost looks like nothing is there, which is surprising and not totally expected. Measuring these samples at high magnification is needed to support the arguments.

Answer: Thanks a lot for your comments. According to the reviewer's comments, we have re-measured these samples at high magnification SEM ($\times 5k$, $\times 10k$ and $\times 20k$) in the manuscript (Fig. 1 h-1/h-2) and supporting information (Fig. S50). We have included the updated results and corresponding comments in the revised manuscript.

Fig. S50 (A) SEM image of PPE-Ni₂-MgO/Ni₃-MgO (prepared by mixed catalyst Ni₂-MgO and Ni₃-MgO). (B) SEM image of PPE-Ni₂/Ni₃-MgO (prepared by co-anchored heterogeneous catalyst Ni₂/Ni₃-MgO).

These polymers were melt-pressed at about 150 °C, 5 MPa for 5 min to obtain the test specimen, and freeze it with liquid nitrogen and then wetting-off to obtain the fracture surface for SEM observation. As shown in Fig. S50A and B, obviously, the SEM image of polar bimodal polyethylene PPE-Ni₂-MgO/Ni₃-MgO indicated the presence of more phase separation than PPE-Ni₂/Ni₃-MgO.

The figure in the manuscript was modified as follows,

Fig. 1 Comparison of bimodal polyethylene prepared in different systems. (a) Tensile curves of polyethylene prepared by homogeneous polymerization. PE-Ni1 (Table S2, Entry 1), PE-Ni3 (Table S2, Entry 3), PE-Ni1/Ni3 (Table S2, Entry 17). (b) Tensile curves of polyethylene prepared by heterogeneous polymerization. PE-Ni2-MgO (Table S2, Entry 5), PE-Ni3-MgO (Table S2, Entry 6), PE-Ni2/Ni3-MgO (Table S2, Entry 16), PE-Ni2-MgO/Ni3-MgO (Table S2, Entry 18). (c) Tensile curves of polar polyethylene prepared by homogeneous polymerization. PPE-Ni1 (Table S3, Entry 1), PPE-Ni3 (Table S3, Entry 3), and PPE-Ni1/Ni3 (Table S3, Entry 4). (d) Tensile curves of polar polyethylene prepared by heterogeneous polymerization. PPE-Ni2-MgO (Table 1, Entry 5), PPE-Ni3-MgO (Table 1, Entry 6), PPE-Ni2/Ni3-MgO (Table 1, Entry 12), and PPE-Ni2-MgO/Ni3-MgO (Table 1, Entry 22). (e) Repetitive tensile curves of polymers PE-Ni2/Ni3-MgO and PE-Ni2-MgO/Ni3-MgO. (f) DSC curve of PPE-Ni2-MgO, PPE-Ni3-MgO, PPE-Ni2-MgO/Ni3-MgO and PPE-Ni2/Ni3-MgO. (g) Rheological curve of PPE-Ni2-MgO/Ni3-MgO and PPE-Ni2/Ni3-MgO. (h) h-1, SEM image of PPE-Ni2-MgO/Ni3-MgO. h-2, SEM image of PPE-Ni2/Ni3-MgO. The original SEM images were listed in Fig. S50.

Somewhat related, the 3D printing data is compelling, and I wonder if SEM data for products in Fig. 2H and 2I would be a more compelling proof-of-concept experiment showing how phase segregation does (or does not) occur with the PPE materials.

Answer: Thanks a lot for your comments. According to the reviewer's comments, we have measured SEM of these samples in the supporting information (Fig. S55).

Fig. S55 SEM data for products in Fig. 4H and 4I in the manuscript. (a) and (b), SEM for HDPE: PLA 7:3 commercial (prepared by blending HDPE and polylactic acid in a ratio of 7 to 3). (c) and (d), SEM for PPE-**Ni2/Ni3-APP**: PLA 7:3 (prepared by PPE-**Ni2/Ni3-APP** and polylactic acid in a ratio of 7 to 3).

We have revised the description in the manuscript,

“The good surface properties of these polar bimodal polyethylene samples also enabled good compatibility with other types of polymers, making them more versatile for tuning material properties. For example, as shown in Fig. S55, after blending non-polar HDPE with polylactic acid (PLA), the SEM image showed obvious "sea-island" phase separation. However, the polar bimodal polyethylene PPE-**Ni1/Ni3-APP** with polar groups showed excellent polar compatibility, therefore, it was much easier to 3D print blends of PLA with polar bimodal polyethylene versus commercial HDPE (Fig. 4h vs 4i).”

Reporting images of extrudate of PPE with mixtures of Ni2/MgO and Ni3/MgO should be given. Also, the images are not particularly clear, it was not obvious to me what I should be looking at in these images.

Answer: Thanks a lot for your comments. According to the reviewer’s comments, we have added the images extrudate of PPE with mixtures of Ni2/MgO and Ni3/MgO (PPE-Ni2-MgO/Ni3-MgO) in the manuscript, and got clearer pictures as follows.

Fig. S53 Images of extruded samples of PE-Ni₂-MgO (e1, Table S1, Entry 5), PPE-Ni₂-MgO/Ni₃-MgO (e2, Table 1, Entry 22) and PPE-Ni₂/Ni₃-MgO (e3, Table 1, Entry 12).

We have revised the description in the manuscript,

“Clearly, the extruded polar bimodal polyethylene sample (e2 and e3) was much smoother and more uniform than the sample with non-polar polyethylene (e1, Table S2, entry 5, PE-Ni₂-MgO). The introduction of polar functional groups and the presence of low-molecular-weight copolymers may have both contributed to its good extrusion performance. The extrusion performance of polar bimodal polyethylene prepared by co-anchoring strategy (e3, Table 1, Entry 12, PPE-Ni₂/Ni₃-MgO) was better than that from separately mixed supported heterogeneous catalyst (e2, Table 1, Entry 22, PPE-Ni₂-MgO/Ni₃-MgO), which further indicated that the two components of polar bimodal polyethylene prepared by co-anchoring strategy had better blending properties.”

A stylistic point, but one the authors should consider revising: many of the properties discussed on page 5 (dyeing, O₂ permeability, and 3D printing) are written like a progress report. In addition, more description of how the gas barrier properties are measured would be helpful to the general audience.

Answer: Thanks a lot for your comments. According to the reviewer’s comments, we have revised the relevant description in the manuscript.

“Dyeing Properties. The dyeing properties of polyolefins are related to their surface properties.^{40,41} The polar bimodal polyethylene and dye powder (2,2'-[(3,3'-Dichloro[1,1'-biphenyl]-4,4'-diyl)bis(azo)]bis[N-(2-methylphenyl)-3-oxobutamide]) were mixed. The blends were melt-pressed at 150 °C to obtain test specimens (Fig. 4c). Subsequently, the specimens were washed with hot acetone for 72 h and dried in a vacuum oven to a constant weight. In the UV-vis absorption spectra, the absorbance peak at 465 nm for polar bimodal polyethylene only decreased slightly after washing (Table 1, Entry 9). In contrast, the unimodal

polar polyethylene with the same comonomer content (Table 1, Entry 7) showed a much more dramatic decrease. This is consistent with the surface property studies using WCAs and indicates the superior surface properties of the polar bimodal polyethylene prepared using the co-anchoring strategy, which further indicates that the surface of the film prepared by melting processing may contain more polar functional groups.

Gas Barrier Properties. The gas barrier properties of polymer films are very important for applications such as food packaging. After melting and pressing polar bimodal polyethylene and unimodal polyethylene prepared by different catalyst systems into thin films, we use GAS PERMEABILITY TESTER to measure the barrier of the film to oxygen. The results showed that the oxygen barrier properties of polar bimodal polyethylene prepared by co-anchored catalysts were even better than those of the unimodal polymer sample (Fig.4d; 3.22 vs 1.41, 3.84 vs 1.62). It is possible that the low-molecular-weight fraction, especially its good miscibility and entanglement with the high-molecular-weight fraction, made the film denser and improved its oxygen barrier properties. Furthermore, the bimodal copolymer prepared by separately mixing supported heterogeneous catalyst **Ni2-MgO/Ni3-MgO** showed poor oxygen barrier performance (5.02). This may be due to great phase compatibility of the polar bimodal polyethylene prepared by co-anchored catalyst (as reflected by SEM, Fig. 1h), leading to high molecular chain entanglement and the formation of dense film.

Extruding Properties. Bimodal polyethylene can improve processability.⁷ The extrusion properties of polar bimodal polyethylene samples were studied with a single-screw extruder at 200 °C (Fig. 4e). Clearly, the extruded polar bimodal polyethylene sample (e2 and e3) was much smoother and more uniform than the sample with non-polar polyethylene (e1, Table S2, entry 5, PE-**Ni2-MgO**). The introduction of polar functional groups and the presence of low-molecular-weight copolymers may have both contributed to its good extrusion performance. The extrusion performance of polar bimodal polyethylene prepared by co-anchoring strategy (e3, Table 1, Entry 12, PPE-**Ni2/Ni3-MgO**) was better than that from separately mixing supported heterogeneous catalyst (e2, Table 1, Entry 22, PPE-**Ni2-MgO/Ni3-MgO**), which further indicated that the two components of polar bimodal polyethylene prepared by co-anchoring strategy had better blending properties. The introduction of polar functional groups and the presence of low-molecular-weight copolymers may have both contributed to its good extrusion performance.

3D Printing. The emergence of three-dimensional (3D) printing has added a new dimension to polymer processing and holds huge prospects for manufacturing complex multi-functional material systems in a single processing step.^{42,43} However, 3D printing high-density polyethylene (HDPE) has been problematic owing to its massive shrinkage, accompanied by its poor adhesion to common build plates.⁴⁴ Thus, it is difficult to 3D print commercial HDPE (Fig. 4f). The polar bimodal polyethylene material enabled by the co-anchoring strategy showed both improved extrusion properties and good surface properties. These properties made 3D printing viable for these polar polyethylene materials (Fig. 4g). The deliberately-chosen APP support also rendered the material flame retardant (Table 1, entry 19). The good surface properties of these polar bimodal polyethylene samples also enabled good compatibility with other types of polymers, making them more versatile for tuning material properties. For example, as shown in Fig. S55, after blending non-polar HDPE with polylactic acid (PLA), the

SEM image showed obvious "sea-island" phase separation. However, the polar bimodal polyethylene PPE-Ni1/Ni3-APP with polar groups showed excellent polar compatibility, therefore, it was much easier to 3D print blends of PLA with polar bimodal polyethylene versus commercial HDPE (Fig. 4h vs 4i)."

We modified the description of oxygen barrier performance test in the Methods of manuscript,

"Gas barrier experiment. Oxygen permeability was determined by GAS PERMEABILITY TESTER (Basic 201, Jinan Languang Electromechanical Technology Co., Ltd) according to GB/T 1038-2000 at 25°C. The polymer was melted and pressed at about 150 °C, 5MPa for 10 minutes to obtain a film with the thickness of about 0.5 mm, and cut it into a circle with the diameter of 10 mm for gas barrier experiment. Install the prepared film into the vacuum chamber of the gas permeability tester, inject oxygen, measure the permeability of the film after a certain time, and prepare at least three samples of each polymer for testing."

Several minor typos in the need to be corrected (D6H6, O2 barrea, etc)

Answer: Thanks a lot for your comments. According to the reviewer's comments, we have corrected the typos in the manuscript and supporting information.

Reviewer #2 (Remarks to the Author):

The manuscript "A Co-Anchoring Strategy for the Synthesis of Polar Bimodal Polyethylene" describes the co-anchoring of Ni olefin polymerization catalysts to MgO for the purposes of making blends of high MW HDPE (made by catalyst 1 or 2) with lower Mw functionalized PE (made by catalyst 3). The performance of the supported catalysts is better than the homogeneous analogs, and co-anchored mixtures is better than mixtures of the individual heterogeneous catalysts. The tensile properties, rheology, melting behavior, wetting studies, oxygen permeability, and 3D printing results were reported and support the hypothesis that the polar functionality enhances the material properties of the PE blends.

I have some questions about catalyst 3, however. The authors report that a complex with the formula (L)Ni(COD) is formed where L is the mono-deprotonated ligand; however, not enough information is provided. As written, the complex is formally Ni(I) and would be paramagnetic, but NMR data is reported that is not consistent with this hypothesis. I would expect that either a Ni(0) complex (LHNiCOD) or Ni(II) complex (LNiCODH) would form, where CODH is bound to Ni as an alkyl or allylic substituent. I have drawn some of these in the attached file. The NMR data are listed but not assigned, and the spectrum is not shown in the SI. Likewise the elemental analysis (both calculated and experimentally observed) was listed for the LNi fragment (C₂₂H₂₂NiO₂P) rather than LNi(COD) (C₃₀H₃₅NiO₂P). It is also unclear why Ni(COD)₂ was used as a precursor rather than (py)₂NiMe₂, which was used for complexes 1 and 2, or other Ni(II) precursors such as Ni(allyl)Br dimer, py₂Ni(CH₂TMS)₂, (PR₃)₂NiPhCl, etc.

Although the syntheses of the catalysts is not really the focus of the paper, the lack of characterization for catalyst 3, or even a discussion of previous studies are concerning given that catalyst 3 is the one making the functionalized polymer. For this reason, I cannot

recommend this manuscript for publication. I will be happy to re-review it once these questions are addressed.

Answer: Thanks a lot for your comments. According to the reviewer's comments, we have obtained high quality characterization Ni₃ catalyst and included the details in the revised manuscript and supporting information.

Our initial attempt to react ligand **L3** with (Py)₂NiMe₂ failed to give any isolable product. It may be due to the side reaction of this nickel precursor with the para-hydroxy group of ligand **L3** with small steric hindrance. So we employed an alternative synthetic strategy for Ni₃ according to the reference (J. Am. Chem. Soc. 2017, 139, 3611-3614). We have added corresponding characterization in the support information.

We have redrawn the structure of catalyst Ni₃ and carefully characterized it as follows.

Fig. S67 ¹H NMR spectrum of the Ni **3**. (C₆D₆, 400 MHz)

Fig. S68 ^{31}P NMR spectrum of the Ni **3**. (C_6D_6 , 400 162 MHz)

Fig. S69 ^{13}C NMR spectrum of the Ni **3**. (C_6D_6 , 400 101MHz)

Ni3: ^1H NMR (400 MHz, C_6D_6) δ 7.31 (ddd, $J = 30.1, 10.9, 7.2$ Hz, 4H), 6.91- 6.65 (m, 5H), 6.59(s, 1H), 6.45 (s, 1H), 6.20 - 6.07 (m, 1H), 5.66 -5.12 (m, 2H, $-\text{CH}=\text{CH}-$), 3.76 (s, 1H, $-\text{OH}$), 2.09 - 1.79 (m, 4H), 1.74 (q, 1H), 1.70 -1.60 (m, 1H), 1.56 (s, 1H), 1.45-1.39 (m,1H), 1.34 (s, 9H, $-t\text{Bu}$), 1.19 - 1.07 (m, 2H), 0.80 - 0.69 (m, 1H); ^{31}P NMR (162 MHz, C_6D_6) δ 16.96. ^{13}C NMR (101 MHz, C_6D_6) δ 145.31, 138.47, 133.11, 132.25, 132.12, 130.17, 128.53, 127.64,

Anal. Calcd for C₃₀H₃₅NiO₂P: C, 69.66; H, 6.82. Found: C, 69.52; H, 6.89.

Reviewer #3 (Remarks to the Author):

Zou and coworkers report a strategy for the synthesis of blends of two polymers: 1) a low MW fraction with high levels of polarity, and 2) a higher MW fraction with fewer polar groups. It is proposed that such a mixture will have the ideal properties of good mechanical properties, along with beneficial properties (dyeing, gas barrier, extrusion printing, etc). To achieve miscibility between these two components, three synthetic routes were explored: mixtures of homogeneous catalysts, separately supported heterogeneous catalysts, and a co-anchoring strategy. It was claimed that the co-anchoring strategy worked better than the other two strategies.

First, I feel that this paper addresses an important topic, and that the science here is excellent. However, I believe that the paper is fairly applied, and will likely be of interest to a select group of scientists working in the area of functional polyolefins, rather than a broad scientific audience. For these reasons, it is my opinion that this work would be better suited to a more specialized journal focusing on polymer synthesis. I would consider this for Nature Communications if the work were less empirical. For example, it is unclear to me how different the levels of functionality can be while still achieving miscibility. I would assume that at some gap in functionality, that the materials would phase separate. If the authors could make an array of PE materials with varying levels of functional group incorporation using single catalysts under controlled conditions, then map out the phase space for miscibility or phase separation, I would view this to significantly improve the scientific component of the paper. As it stands, I feel it would be better suited for a more specialized polymer journal.

Answer: Thanks a lot for your comments. According to the reviewer's comments, we have supplemented the following experiments and discussions, including the study of the effects of different polar functional groups on the phase composition, mechanical properties, rheological properties, etc., in order to improve the scientific component of the manuscript.

1. We have prepared a series of bimodal polyethylene with different comonomer incorporation ratios using co-anchored catalyst and mixed heterogeneous catalyst, and compared the phase compatibility, mechanical properties and rheological properties of these polyethylene, as shown in Table S4, Fig. 2, Fig. 3 and Fig. S51. In addition to the supported system of MgO, the phase separation behavior of the supported system of ammonium polyphosphate (APP) was also studied in Fig. S52.

Table S4. Synthesis of a series of polar bimodal polyethylene. ^a

Ent.	Cat.	Mon. (mol/L)	Yield/ g ^b	Act. b(10 ⁵)	X _M ^c (%)	T _m ^d /°C	M _n e(10 ⁴)	M _w e(10 ⁴)	M _w /M _n ^e
1	Ni2/Ni3-MgO(1:1)	-	0.43	258.0	0	133.6	3.0	108.9	35.8
2	Ni2/Ni3-MgO(1:1)	0.5	0.62	12.4	0.5	132.2	6.8	78.9	11.7
3	Ni2/Ni3-MgO(1:5)	0.5	0.33	6.6	0.9	129.1	4.0	52.6	13.3

4 ^f	Ni2/Ni3-MgO(1:1)	0.5	0.58	11.6	1.3	126.4	4.7	17.4	3.7
5	Ni2/Ni3-MgO(1:5)	1.0	0.39		1.7	121.5	0.7	7.3	10.0
6	Ni2-MgO/Ni3-MgO(1:1)	-	0.43	276.0	0	125.9/ 135.8	7.0	117.1	16.7
7	Ni2-MgO/Ni3-MgO(1:1)	0.5	0.52	1.04	0.5	125.6/ 131.1	2.1	72.7	32.4
8	Ni2-MgO/Ni3-MgO(1:5)	0.5	0.35	7.0	0.9	122.8/1 26.6	1.0	57.5	55.2
9 ^f	Ni2-MgO/Ni3-MgO(1:1)	0.5	0.47	9.4	1.3	116.5/1 28.2	1.28	14.5	11.3
10	Ni2-MgO/Ni3-MgO(1:5)	1.0	0.36	7.2	1.7	109.3/1 22.91	0.8	7.6	9.9

^a Conditions: cat. 1 μmol (Ni); 5 mL Hex.; $t = 30$ min; $T = 80$ $^{\circ}\text{C}$; $P_{\text{ethylene}} = 8$ atm. Comonomer: undecenoic acid. ^b Yields are the average of at least two runs. Activity is in units of 10^5 g/(mol cat. \times h). ^c Incorporation ratios of comonomers were determined from ^1H NMR spectra. ^d Determined by differential scanning calorimetry (DSC, second heating) ^e M_n : 10^4 g mol $^{-1}$, M_n , M_w , and M_w/M_n were determined by gel permeation chromatography in 1,2,4-trichlorobenzene at 160 $^{\circ}\text{C}$. ^f $T = 120$ $^{\circ}\text{C}$, 8 atm.

Fig. 2 Comparison of SEM images of two polar bimodal polyethylene prepared by co-anchored catalyst and mixed heterogeneous catalyst after incorporation of polar monomer. The upper polymers were prepared by co-anchoring strategy, and the lower polymers were prepared by mixed heterogeneous catalyst.

Fig. 3 Correlation diagram of tensile strength and toughness with polar monomer incorporation of a series of bimodal polyethylene.

Fig. S51 Tensile curve (a) and complex viscosity curve (b) of a series of bimodal polyethylene.

Fig. S52 SEM of a series of bimodal polyethylene prepared by APP supported catalyst. The upper polymers were prepared by co-anchoring strategy, and the lower polymers were prepared by mixed heterogeneous catalyst.

We have added the following discussions to the manuscript,

“We have prepared a series of bimodal polyethylene samples with different comonomer incorporation ratios using co-anchored catalyst and mixed heterogeneous catalyst (Table S4), and compared their phase compatibility, mechanical properties and rheological properties. At low comonomer incorporation ratio (<0.5%), SEM images showed uniform homogeneity for both cases (Fig. 2). However, the samples prepared by mixed heterogeneous catalyst **Ni2-MgO/Ni3-MgO** showed obvious phase separation at incorporation ratios of above 0.9% (Fig. 2). In direct contrast, the samples prepared using co-anchored catalyst **Ni2-Ni3/MgO** maintained great compatibility even at high comonomer incorporation (1.7%).

Similar with the SEM results, the mechanical properties of the samples prepared by co-anchoring strategy were only slightly decreased with increasing comonomer incorporation (0-1.7%) (Fig. 3, Fig. S51). However, the samples prepared by mixed heterogeneous catalyst showed extremely poor mechanical properties at comonomer incorporation ratio of above 0.5%, due to the obvious phase separation of the two components. In particular, the toughness of the material decreases sharply and almost disappears after the introduction of polar monomer. Clearly, the mechanical properties of bimodal polymer samples prepared by the two catalyst

systems are quite different due to the differences of their microscopic phase separation behaviors.

The comparison of rheological properties of these samples also showed that the complex viscosity of bimodal polyethylene prepared by co-anchoring strategy were much higher than those prepared by mixed heterogeneous catalyst before the melting temperature, indicating that the two components of bimodal polyethylene prepared by co-anchoring strategy are more entangled. In addition, similar results were observed for other types of supported heterogeneous catalysts (APP) (Fig. S52).”

2. According to the reviewer's comments, in order to improve the scientific content of the manuscript, we measured the SEM of these 3D printing samples in the supporting information to further study and verify the compatibility of polar bimodal polyethylene prepared by anchoring strategy with polar polymer (such as PLA) and the 3D printing performance of their blends.

Fig. S55 SEM data for products in Fig. 4H and 4I in the manuscript. (a) and (b), SEM for HDPE: PLA 7:3 commercial (prepared by blending HDPE and polylactic acid in a ratio of 7 to 3). (c) and (d), SEM for PPE-Ni2/Ni3-APP: PLA 7:3 (prepared by PPE-Ni2/Ni3-APP and polylactic acid in a ratio of 7 to 3).

We have revised the description in the manuscript,

“The good surface properties of these polar bimodal polyethylene samples also enabled good compatibility with other types of polymers, making them more versatile for tuning material properties. For example, as shown in Fig. S55, after blending non-polar HDPE with polylactic acid (PLA), the SEM image showed obvious "sea-island" phase separation. However, the polar bimodal polyethylene PPE-Ni1/Ni3-APP with polar groups showed excellent polar compatibility, therefore, it was much easier to 3D print blends of PLA with polar bimodal polyethylene versus commercial HDPE (Fig. 4h vs 4i).”

3. According to the reviewer's comments, in order to improve the scientific composition of the manuscript, we further studied the extrusion performance of polar bimodal polyethylene, and

added the image extrusion of the mixture of PPE and Ni₂/MgO and Ni₃/MgO (PPE-Ni₂-MgO/Ni₃-MgO) to the manuscript, which was compared with the polar polyethylene prepared by anchoring strategy.

Fig. S53. Images of extruded samples of PE-Ni₂-MgO (e1, Table S1, Entry 5), PPE-Ni₂-MgO/Ni₃-MgO (e2, Table 1, Entry 22) and PPE-Ni₂/Ni₃-MgO (e3, Table 1, Entry 12).

We have revised the description in the manuscript,

“Clearly, the extruded polar bimodal polyethylene sample (e2 and e3) was much smoother and more uniform than the sample with non-polar polyethylene (e1, Table S2, entry 5, PE-Ni₂-MgO). The introduction of polar functional groups and the presence of low-molecular-weight copolymers may have both contributed to its good extrusion performance. The extrusion performance of polar bimodal polyethylene prepared by co-anchoring strategy (e3, Table 1, Entry 12, PPE-Ni₂/Ni₃-MgO) was better than that of prepared by separately mixing supported heterogeneous catalyst (e2, Table 1, Entry 22, PPE-Ni₂-MgO/Ni₃-MgO), which further indicated that the two components of polar bimodal polyethylene prepared by co-anchoring strategy had better blending properties.”

As an added note, I believe Ref 34 is incorrect.

Answer: Thanks a lot for your comments. According to the reviewer’s comments, we have corrected the Ref 34 in the manuscript.

Reviewers' Comments:

Reviewer #1:

Remarks to the Author:

Overall the authors handled most of the comments from the previous round of review. I have only one minor comment regarding Table 1 that should be handled before accepting the article. I brought up in the last round that the values for bimodal PPE are incorrect because this is really a bimodal mixture of polymers, not a very broad molecular weight distribution. The authors responded by fitting the GPC data to two narrow molecular weight fractions, and the data presented in the revision appears correct, and is compelling. However, they put this data in the SI, which I feel is incorrect. Table S2 should replace Table 1 that is currently shown in the text; Table 1 is not necessary in the SI nor the text with the information in current Table S2. They may also consider showing a representative GPC trace in the main text as well, but I leave that up to them to decide.

Reviewer #2:

Remarks to the Author:

The authors have answered the concerns I mentioned in my last review.

Reviewer #3:

Remarks to the Author:

The authors have satisfied my original concerns, and I am in favor of publication.

Reviewer #1 (Remarks to the Author):

Overall the authors handled most of the comments from the previous round of review. I have only one minor comment regarding Table 1 that should be handled before accepting the article. I brought up in the last round that the values for bimodal PPE are incorrect because this is really a bimodal mixture of polymers, not a very broad molecular weight distribution. The authors responded by fitting the GPC data to two narrow molecular weight fractions, and the data presented in the revision appears correct, and is compelling. However, they put this data in the SI, which I feel is incorrect. Table S2 should replace Table 1 that is currently shown in the text; Table 1 is not necessary in the SI nor the text with the information in current Table S2. They may also consider showing a representative GPC trace in the main text as well, but I leave that up to them to decide.

Answer: Thanks a lot for your comments. According to the reviewer's comments, we have listed the fitting GPC data in Table 1.

Reviewer #2 (Remarks to the Author):

The authors have answered the concerns I mentioned in my last review.

Answer: Thank you very much for your comments and support.

Reviewer #3 (Remarks to the Author):

The authors have satisfied my original concerns, and I am in favor of publication.

Answer: Thank you very much for your comments and support.